# Plasticity-dependent, full detonation at hippocampal mossy fiber–CA3 pyramidal neuron synapses

**Nicholas P Vyleta[1,2], Carolina Borges-Merjane[1], Peter Jonas[1]\***

[1]Institute of Science and Technology Austria, Klosterneuburg, Austria; [2]Vollum Institute, Oregon Health and Science University, Portland, United States

**Abstract** Mossy fiber synapses on CA3 pyramidal cells are 'conditional detonators' that reliably discharge postsynaptic targets. The 'conditional' nature implies that burst activity in dentate gyrus granule cells is required for detonation. Whether single unitary excitatory postsynaptic potentials (EPSPs) trigger spikes in CA3 neurons remains unknown. Mossy fiber synapses exhibit both pronounced short-term facilitation and uniquely large post-tetanic potentiation (PTP). We tested whether PTP could convert mossy fiber synapses from subdetonator into detonator mode, using a recently developed method to selectively and noninvasively stimulate individual presynaptic terminals in rat brain slices. Unitary EPSPs failed to initiate a spike in CA3 neurons under control conditions, but reliably discharged them after induction of presynaptic short-term plasticity. Remarkably, PTP switched mossy fiber synapses into full detonators for tens of seconds. Plasticity-dependent detonation may be critical for efficient coding, storage, and recall of information in the granule cell–CA3 cell network.

**\*For correspondence:** peter.
jonas@ist.ac.at

**Competing interests:** The authors declare that no competing interests exist.

## Introduction

The strength of a synapse in relation to action potential threshold is a key factor that defines its role in neuronal information processing. In neocortical and hippocampal circuits, it is generally thought that individual synapses are weak, and that temporal or spatial summation of unitary excitatory post-synaptic potentials (EPSPs) is required to reach the firing threshold (*London and Häusser, 2005*). In contrast, in the peripheral nervous system and brainstem, 'detonator' synapses are abundant, which directly control the activity of postsynaptic targets in the absence of temporal and spatial summation. Well established examples of detonator synapses include the neuromuscular junction (*Katz, 1969*) and the calyx of Held in the auditory brainstem (*Borst et al., 1995*). Whether detonator synapses exist in higher brain regions is presently unclear. Resolving this question has important implications for understanding the nature of information encoding and the rules of synaptic computation (*Koch and Segev, 2000*; *London and Häusser, 2005*; *Silver, 2010*). In a network with weak synapses below detonation, activity in an ensemble of multiple convergent presynaptic neurons is required to activate a postsynaptic cell. In contrast, in a network with strong synapses near detonation, activity of a single connected presynaptic neuron may be sufficient to drive a postsynaptic cell and to initiate a behaviorally relevant output (*Brecht et al., 2004*).

The hippocampal mossy fiber synapse onto CA3 pyramidal neurons is a candidate for a detonator synapse (*Henze et al., 2002*; *Pelkey and McBain, 2005*; *Urban et al., 2001*). However, the efficacy of this key hippocampal synapse has not been quantified. Structural properties suggested that mossy fiber synapses are highly efficient, because of proximal dendritic location, multiple release sites, and large vesicular pool (*Acsády et al., 1998*; *Chicurel and Harris, 1992*; *Rollenhagen et al., 2007*). In contrast, functional analysis *in vitro* indicated that mossy fiber synapses have a low release

probability (*Lawrence et al., 2004*; *Vyleta and Jonas, 2014*) and that a single mossy fiber EPSP is insufficient to drive postsynaptic CA3 pyramidal cells (*Brown and Johnston, 1983*; *Jonas et al., 1993*; *Kim et al., 2012*; *Mori et al., 2004*). However, *in vitro* analysis also revealed that mossy fiber synapses show substantial facilitation (*Salin et al., 1996*; *Toth et al., 2000*). Furthermore, *in vivo* experiments demonstrated that high-frequency firing in granule cells and subsequent temporal summation initiates spikes in target pyramidal cells (*Henze et al., 2002*). This has led to the hypothesis that the mossy fiber synapse operates as a 'conditional detonator' (*Henze et al., 2002*; *Pelkey and McBain, 2005*; *Urban et al., 2001*). Alternatively, spatial summation of inputs from multiple mossy fiber synapses, or spatial summation of mossy fiber and non-mossy fiber inputs may be required for spike initiation (*Henze et al., 2002*; *Pelkey and McBain, 2005*). Whether single unitary mossy fiber EPSPs trigger spikes in CA3 pyramidal neurons in the absence of summation is currently unknown.

Mossy fiber synapses not only exhibit facilitation, but uniquely large and prolonged post-tetanic potentiation (PTP) (*Griffith, 1990*; *Langdon et al., 1995*; *Salin et al., 1996*; *Zucker and Regehr, 2002*). Here, we tested whether PTP could convert mossy fiber synapses from subdetonator into full detonator mode. The efficacy of mossy fiber synapses has been hard to tackle, because it is difficult to selectively stimulate mossy fibers, and, in particular, to selectively activate single presynaptic terminals (*Nicoll and Schmitz, 2005*). We address this question with a recently developed method (*Vyleta and Jonas, 2014*) to selectively and minimally invasively stimulate an individual presynaptic terminal, while simultaneously recording from an adjacent postsynaptic CA3 pyramidal neuron.

## Results

To determine the efficacy of individual mossy fiber connections, we used our recently developed technique to selectively and noninvasively stimulate single mossy fiber presynaptic terminals (*Figure 1*) (*Vyleta and Jonas, 2014*). Single mossy fiber terminals were stimulated in the tight-seal, bouton-attached configuration (*Figure 1A–C*), while postsynaptic CA3 pyramidal neurons were simultaneously recorded in the whole-cell configuration at near-physiological temperature (31–34°C) (*Alcami et al., 2012*; *Bischofberger et al., 2006*; *Perkins, 2006*; *Vyleta and Jonas, 2014*). Bouton-attached stimulation enabled precise control of electrical activity, as verified by the appearance of action currents in the presynaptic terminal (*Figure 1C*). Recently, this method was also successfully used at the calyx of Held (*Acuna et al., 2015*). Hippocampal granule cells *in vivo* fire both single action potentials and bursts of spikes (*Pernía-Andrade and Jonas, 2014*). We first examined the properties of synaptic transmission evoked by single presynaptic action potentials at mossy fiber–CA3 pyramidal neuron synapses (*Figure 1D*). Single unitary EPSPs had a mean latency of $1.06 \pm 0.09$ ms, a peak amplitude of $9.6 \pm 2.4$ mV, a 20–80% rise time of $3.6 \pm 0.7$ ms, and a decay time constant of $134 \pm 10$ ms (6–8 pairs; *Figure 1E–I*). Furthermore, single unitary EPSPs triggered action potentials with a mean probability of only $0.12 \pm 0.08$. Thus, single unitary mossy fiber EPSPs failed to reliably induce action potential initiation in postsynaptic CA3 pyramidal neurons.

We next tested whether repetitive stimulation of the presynaptic terminal was sufficient to initiate action potentials in postsynaptic CA3 pyramidal neurons (*Figure 2*). Three brief stimuli were applied at a frequency of 50 Hz, resembling burst activity in hippocampal granule cells *in vivo* (*Pernía-Andrade and Jonas, 2014*). Under these conditions, spike probability for the third EPSP increased to $0.82 \pm 0.09$ (*Figure 2A,B*). The increase in spike probability could be generated by either presynaptic facilitation or postsynaptic summation. To reveal the contribution of presynaptic facilitation (*Salin et al., 1996*; *Toth et al., 2000*; *Vyleta and Jonas, 2014*), we analyzed the underlying postsynaptic conductance changes by measuring excitatory postsynaptic currents (EPSCs) in the same pairs (*Figure 2C-E*). For a 50-Hz train of stimuli, the ratio $EPSC_2 / EPSC_1$ was $2.57 \pm 0.42$, and the ratio $EPSC_3 / EPSC_1$ was $3.12 \pm 0.62$. To assess the contribution of postsynaptic temporal summation, we simulated the extent of superposition of three temporally shifted identical EPSPs. For an EPSP decay time constant of 134 ms, the ratio $EPSP_2 / EPSP_1$ was 1.86, and the ratio $EPSP_3 / EPSP_1$ was 2.60. Thus, the increase of EPSP amplitude during a train was primarily generated by presynaptic facilitation, and to a smaller extent by postsynaptic summation. These results support the idea that the mossy fiber–CA3 pyramidal neuron synapse operates as a 'conditional detonator' (*Henze et al., 2002*), and further identify presynaptic facilitation as the major mechanism underlying conditional detonation.

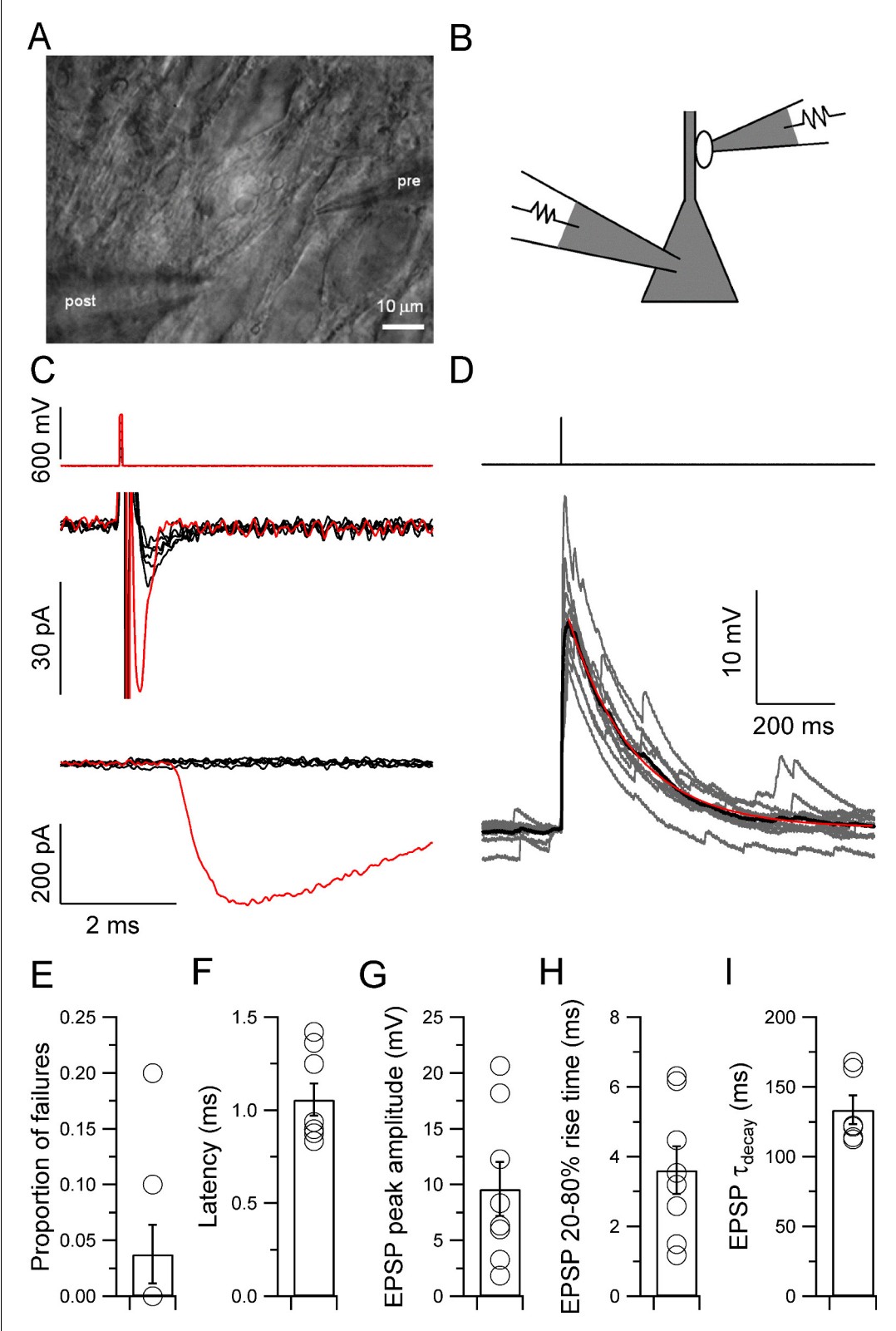

**Figure 1.** Properties of single unitary EPSPs at the hippocampal mossy fiber–CA3 pyramidal neuron synapse. (A, B) Infrared videomicrograph (A) and schematic illustration (B) showing a paired recording from a mossy fiber bouton (tight-seal, bouton-attached configuration) and a synaptically connected CA3 pyramidal neuron (whole-cell configuration). This recording configuration enabled the specific, noninvasive stimulation of a single presynaptic terminal. (C) Stimulation of a presynaptic terminal in the bouton-attached configuration. Top, short pulses of increasing amplitude (50–600

*Figure 1 continued on next page*

*Figure 1 continued*

mV, 50-mV steps); center, presynaptic action currents; bottom, unitary EPSCs. Black, subthreshold; red, suprathreshold stimuli and corresponding responses. Note that the appearance of a presynaptic action current is tightly associated with the generation of a postsynaptic response in the CA3 pyramidal cell. (D) Traces of unitary EPSPs evoked by a single presynaptic stimulus to the mossy fiber bouton (top, 700 mV, 0.1 ms, same recording as in C). Ten consecutive EPSPs are shown superimposed (gray; 20 s repetition interval) overlayed with the average (black). The decay time course of the average EPSP was fit with a monoexponential function (red; decay τ = 166 ms). Note that the slow EPSP decay will promote temporal summation. (E–I) Summary graphs for the proportion of failures (E), latency (F), peak amplitude (G), 20–80% rise time (H), and decay time constant (I) of the average unitary EPSP across recordings (6 to 8 pairs). Circles indicate data from single experiments; bars indicate mean ± SEM.

In addition to pronounced short-term facilitation, mossy fiber synapses also exhibit uniquely large post-tetanic potentiation (PTP) (*Griffith, 1990*; *Langdon et al., 1995*; *Salin et al., 1996*; *Zucker and Regehr, 2002*). Can PTP convert mossy fiber synapses from a subdetonator into a detonator mode? To address this question, we measured the probability of postsynaptic action potential initiation before and after high-frequency stimulation (HFS, 100 stimuli delivered at 100 Hz), a paradigm that reliably induces PTP (*Griffith, 1990*) (*Figure 3*). Intriguingly, HFS substantially increased the probability of spiking for the first stimulus, from $0.14 \pm 0.09$ under control conditions to $0.71 \pm 0.18$ at the peak and to $0.51 \pm 0.16$ in a 100 s time window following HFS (22 to 122 s; seven pairs; p=0.03; *Figure 3D,E*). Detonation was rapid – the latency from the stimulus to the maximum slope of the EPSP preceding action potential generation in the PTP phase was $1.5 \pm 0.09$ ms, and the latency from the stimulus to action potential onset in the PTP phase was only $3.1 \pm 0.09$ ms (seven pairs). Enhanced detonation decayed with a time constant of 67 s, indicating that the alteration was long lasting (*Figure 3D*). HFS also increased the probability of spiking for the second and the third stimulus (*Figure 3E*). In the same paired-recordings, the extent of PTP, quantified from the maximal slope of the rising phase of $EPSP_1$, was 442% at the peak and 348% in the 100 s time interval following HFS (*Figure 3B,C*). PTP decayed with a time constant of 89 s, consistent with a prolonged enhancement of release (*Figure 3B*). Thus, PTP converted mossy fiber synapses from a subdetonator into a full detonator synapse for an extended time period.

## Discussion

The present findings reveal that PTP produces a computational switch at the hippocampal mossy fiber synapse, resulting in full detonation of a CA3 pyramidal neuron in response to a single unitary EPSP. PTP-induced detonation lasts for tens of seconds, consistent with a prolonged enhancement of release from the mossy fiber terminal. Additionally, our results challenge the prevailing view that a single unitary EPSP in the cortex is insufficient to discharge a postsynaptic cell, and that temporal and spatial summation is required to activate the target neuron.

The large extent and the long duration of mossy fiber PTP in comparison to other synapses (*Chu et al., 2014*; *Habets and Borst, 2007*; *Korogod et al., 2005*) suggest that the switch from subdetonation to full detonation may be exquisitely relevant for the function of the hippocampal network *in vivo*. The activity dependence of PTP at different synapses is not precisely known. The amount of PTP appears to depend on both the frequency (*Griffith, 1990*) and the number of action potentials (*Habets and Borst, 2005*), and may also be influenced by other forms of presynaptic plasticity (e.g. frequency facilitation, *Salin et al., 1996*). However, recent *in vivo* whole-cell patch-clamp recordings suggest that granule cells in awake rats and mice occasionally fire bursts of >10 action potentials (*Pernía-Andrade and Jonas, 2014*; *Vyleta et al., 2016*). Furthermore, *in vivo* imaging experiments in awake mice with genetic $Ca^{2+}$ indicators suggest that granule cells occasionally generate large spontaneous $Ca^{2+}$ transients consistent with ~20 action potentials (*Pilz et al., 2016*). Importantly, as granule cell activity seems to be spatially tuned (*Danielson et al., 2016*), even more action potentials may be generated during a long-term stay of the animal in the center of the place field, or if it visits the same place field repeatedly during exploration of the spatial environment (*Danielson et al., 2016*; *Neunuebel and Knierim, 2012*). Therefore, due to the bursting nature of granule cell firing *in vivo*, substantial PTP may be induced during specific behaviors.

The dentate gyrus is also capable of eliciting strong feedforward inhibition onto CA3 pyramidal neurons (*Buzsáki, 1984*), and GABAergic interneurons are a major target for mossy fiber boutons

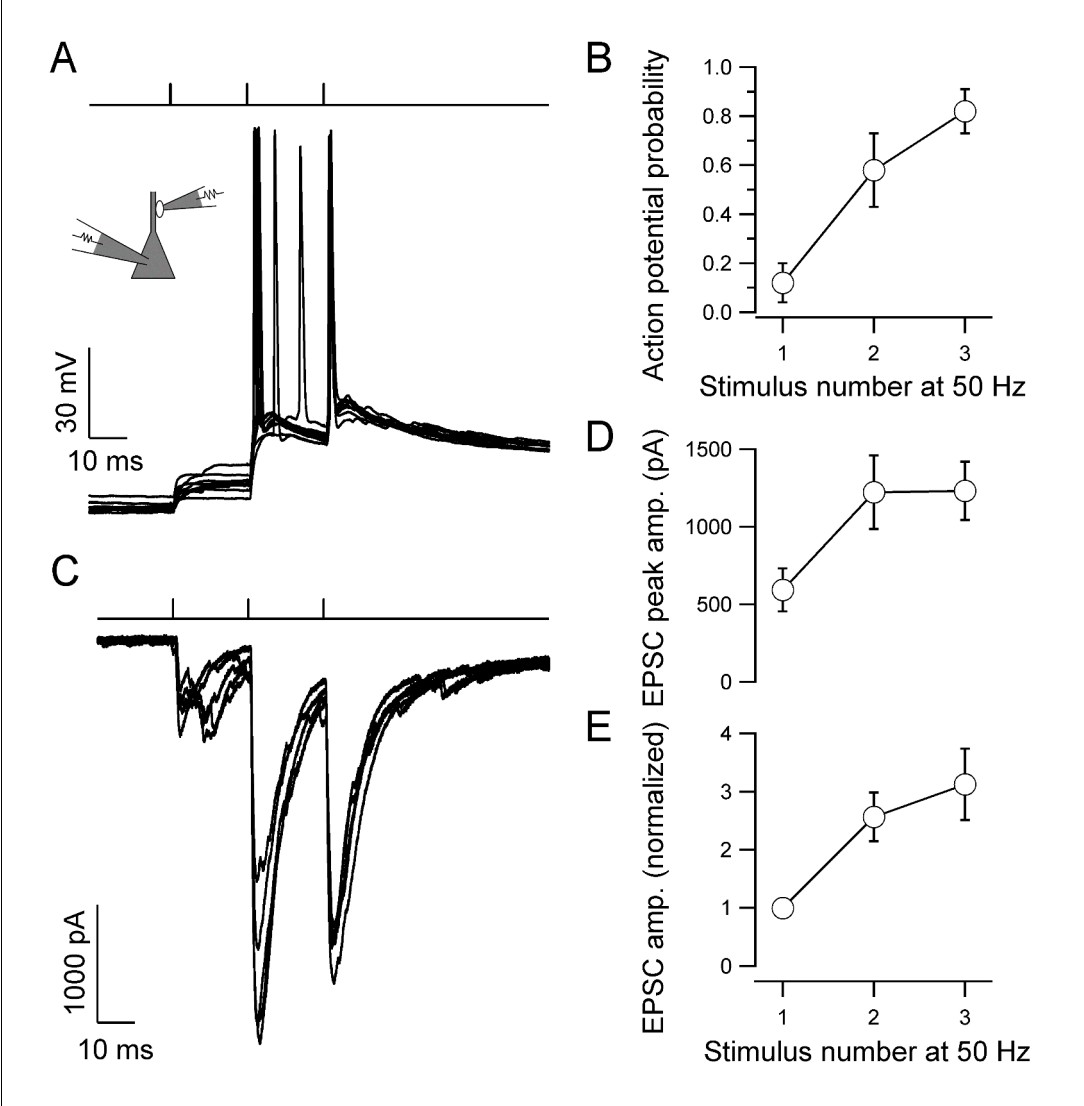

**Figure 2.** Presynaptic facilitation contributes to 'conditional detonation' at the hippocampal mossy fiber–CA3 pyramidal neuron synapse. (A) Pyramidal neuron EPSPs and action potentials evoked by a short train of stimuli delivered to the presynaptic bouton (top, three stimuli delivered at 50 Hz). Ten consecutive postsynaptic responses are shown superimposed (40 s repetition interval). Note that the first presynaptic stimulus was unable to discharge the postsynaptic neuron, whereas the third stimulus reliably initiated firing. (B) Probability of postsynaptic action potential initiation as a function of presynaptic stimulus number (50-Hz stimulation of bouton) for experiments like that shown in A (points connected by lines for clarity, eight pairs). (C) EPSCs recorded in the pyramidal neuron in response to 50-Hz stimulation of the presynaptic bouton, showing marked facilitation of transmitter release (five consecutive EPSCs shown superimposed, 20 s repetition interval, same recording as in A). (D, E) Summary graphs for absolute (D) and normalized EPSC peak amplitude (E) from experiments like that shown in C (13 pairs). Error bars indicate SEM.

(*Acsády et al., 1998*). Although disynaptic inhibition appears to be too slow to affect single action potential generation (*Torborg et al., 2010*) or the rapid detonation that we record here during the PTP phase, inhibition likely prevents excessive depolarization and burst firing of CA3 pyramidal cells (*Torborg et al., 2010*). It should be noted that recruitment of inhibition by cell-attached stimulation of a mossy fiber bouton may be different from that with a physiological stimulus (synaptic activity in granule cell dendrites) for two reasons. First, orthodromic action potential propagation may recruit feedforward inhibition more rapidly and effectively than antidromic action potentials initiated at the mossy fiber terminal. Second, the mossy fiber axon may not be fully maintained in the slice preparation. However, it appears that due to the minimal delay of monosynaptic excitation, large EPSPs

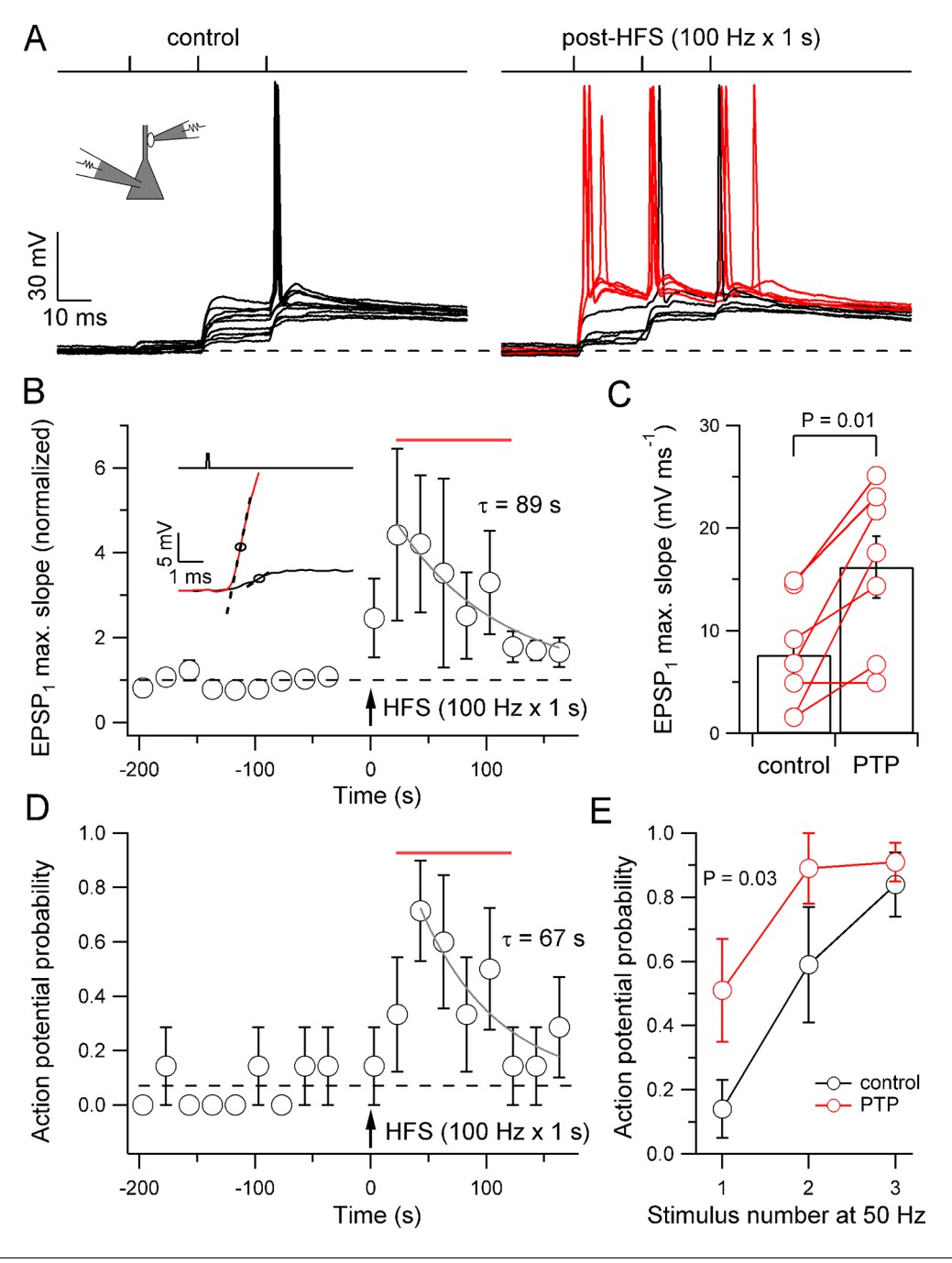

**Figure 3.** PTP converts mossy fiber synapses from a subdetonation into a full detonation mode. (**A**) EPSPs and action potentials in a pyramidal neuron evoked by stimulation of a presynaptic bouton (three stimuli delivered at 50 Hz) before (left, 'control') and after (right, 'post-HFS') a single high-frequency stimulation period (HFS; 100 stimuli delivered at 100 Hz; ten and nine consecutive traces shown superimposed for control and post-HFS, respectively, 20 s repetition interval). After HFS, single presynaptic stimuli were sufficient to discharge the postsynaptic neuron. Red traces were recorded during the 100 s time period defined as PTP (22–122 s after HFS). (**B**) Summary plot of normalized $EPSP_1$ maximum slope versus experimental time for experiments like that shown in **A**. HFS produced an enhancement of EPSP slope which decayed back to baseline (gray, monoexponential curve, $\tau$ = 89 s). Inset: example traces of unitary EPSPs in control (black) and PTP (red, 22 s after HFS) periods from the experiment in **A**. Circles indicate points of maximum slope, dashed lines indicate the corresponding tangential lines (slope, 2.9 and 21.5 mV ms$^{-1}$, respectively). (**C**) Mean $EPSP_1$ maximum slope during control and PTP periods. Open circles connected by lines show data from individual experiments. Bars illustrate mean ± SEM. (**D**) Summary

*Figure 3 continued*

plot of action potential probability during the first stimulus versus experimental time. HFS produced an enhancement of detonation, which slowly decayed back to baseline (gray, monoexponential curve, τ = 67 s). Dashed horizontal lines in **B** and **D** indicate baseline values. (**E**) Probability of action potential initiation in the pyramidal neuron as a function of presynaptic stimulus number (50-Hz stimulation of bouton) for control (black) and PTP (red) periods (seven pairs in **B**–**E**). PTP significantly increased the probability of postsynaptic action potential initiation. Error bars indicate SEM. Time interval used for quantification of PTP effects in C and E is indicated by red horizontal bars in **B** and **D**.

during synaptic enhancement are capable of driving pyramidal cell firing before the effects of disynaptic inhibition.

The results also have major implications for how synaptic computations are performed in the granule cell–CA3 network (*Figure 4*) (*Lisman, 1999*; *Treves and Rolls, 1994*). Current models of hippocampal function suggest that the dentate gyrus processes multimodal sensory inputs by

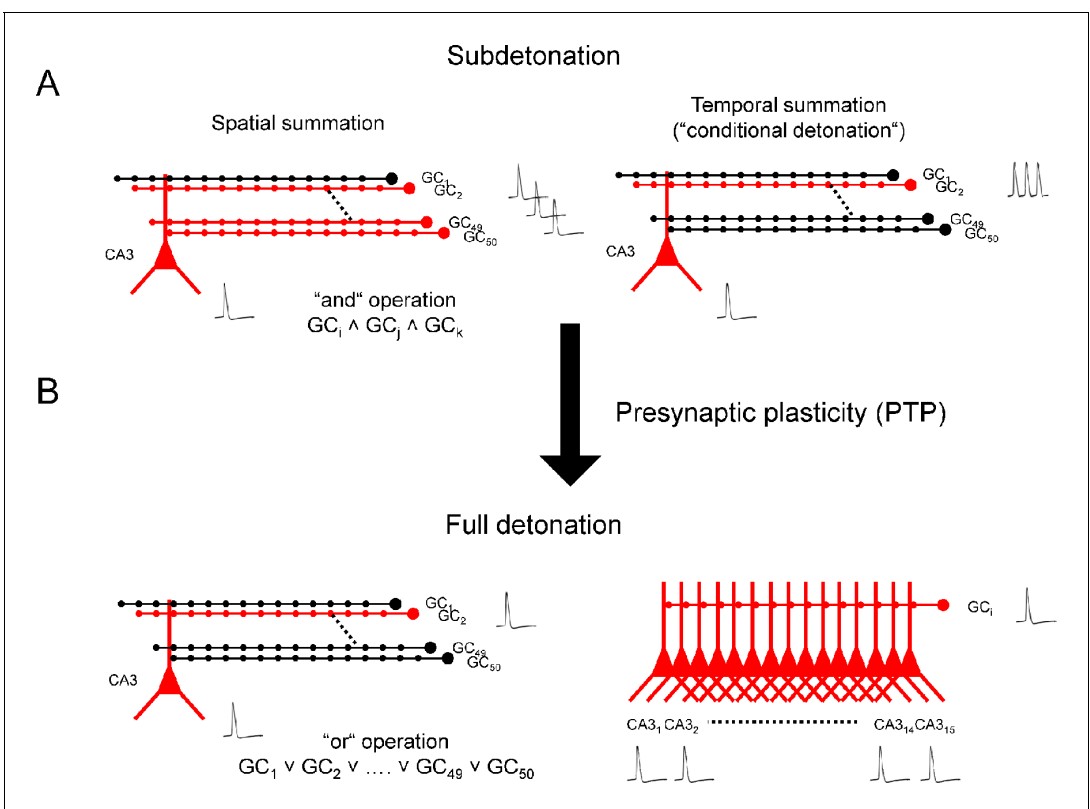

**Figure 4.** Synaptic computations enabled by plasticity-dependent, full detonation in the hippocampal mossy fiber network. (**A**) Synaptic computations in the dentate gyrus–CA3 cell network in a subdetonation regime. Action potential initiation in CA3 pyramidal cells requires spatial summation (e.g. activation of multiple granule cells; left) or temporal summation (e.g. repetitive activation of a single granule cell, right). In the spatial summation scenario, mossy fiber transmission will implement a logic 'and' operation. Specific wiring rules in the network could be generated by structural plasticity (*Galimberti et al., 2006*). Activation of multiple distal perforant path synapses may substitute for the activation of single proximal mossy fiber inputs. (**B**) Synaptic computations in the dentate gyrus–CA3 cell network in a full detonation regime (e.g. during PTP). Action potential initiation in CA3 pyramidal cells is possible after a single action potential in a single granule cell (left). Thus, plasticity-dependent detonation at mossy fiber synapses will enable a logic 'or' operation and allow a single highly specific cue to trigger the encoding, storage, or recall of complex information in CA3 pyramidal neurons (*Quiroga et al., 2005*; *Wilson and McNaughton, 1993*). Additionally, a single spike in a single granule cell may not only activate a single CA3 pyramidal cell, but rather an ensemble of ~15 CA3 cells. Thus, plasticity-dependent detonation at mossy fiber synapses will contribute to the generation of ensemble activity in the hippocampal network. GC, granule cell; CA3, cornu ammonis 3; black, inactive cells; red, active cells. Traces represent single action potentials and action potential trains, respectively.

pattern separation (*Leutgeb et al., 2007*) and represents this information in a sparse coding scheme (*Pernía-Andrade and Jonas, 2014*). A single CA3 pyramidal cell receives convergent input from ~50 granule cells, and a single granule cell provides divergent output onto ~15 CA3 pyramidal neurons (*Amaral et al., 1990*; *Patton and McNaughton, 1995*). How sparse 'engrams' are relayed from the dentate gyrus to the CA3 region has been unclear (*Liu et al., 2012*). In the subdetonation mode, activation of several granule cells by multiple highly specific patterns would be necessary to activate a given CA3 pyramidal neuron (*Figure 4A*). Given the sparse activity of granule cells, this is an unlikely scenario. In contrast, in the full detonation mode, activation of a single granule cell by a single pattern would be sufficient to trigger activation (*Figure 4B*, left). Furthermore, in the full detonation mode, a single action potential in a granule cell should activate an ensemble of ~15 CA3 pyramidal cells (*Figure 4B*, right) (*Amaral et al., 1990*). Thus, our results imply a new mechanism of neuronal ensemble formation in the brain (*Fries, 2009*): divergence combined with plasticity-dependent detonation.

## Materials and methods

### Brain slice preparation

Transverse hippocampal slices (350–400 μm thick) were prepared from 21- to 23-day-old Wistar rats of either sex as described previously (*Bischofberger et al., 2006*). This age range was chosen because morphological properties of mossy fiber synapses were largely mature (*Amaral and Dent, 1981*) and paired recording experiments became increasingly difficult at later stages. Rats were maintained under light (seven am–seven pm) and dark cycle (seven pm–seven am) conditions and were kept in a litter of eight animals together with the mother in a single cage. Animals were killed by rapid decapitation, in accordance with national, institutional, and European guidelines. Experimental procedures were approved by the Bundesministerium für Wissenschaft, Forschung und Wirtschaft (A. Haslinger, Vienna). Slices were cut from the left hemisphere in ice-cold, sucrose-containing extracellular solution using a vibratome (VT1200, Leica Microsystems), incubated in a maintenance chamber at ~34°C for 30–60 min, and subsequently stored at room temperature. Cutting and storage solution contained 87 mM NaCl, 25 mM NaHCO$_3$, 2.5 mM KCl, 1.25 mM NaH$_2$PO$_4$, 10 mM glucose, 75 mM sucrose, 0.5 mM CaCl$_2$, and 7 mM MgCl$_2$, equilibrated with 95% O$_2$ and 5% CO$_2$, ~325 mOsm. Experiments were performed at near-physiological temperature (~32°C; range: 31–34°C).

### Bouton-attached stimulation and paired recording

Subcellular patch-clamp recordings from mossy fiber boutons and simultaneous recordings from pyramidal neurons in the CA3b,c region of the hippocampus were performed under visual control as described previously (*Bischofberger et al., 2006*; *Szabadics and Soltesz, 2009*; *Vyleta and Jonas, 2014*). Slices were superfused with artificial cerebrospinal fluid (ACSF) containing 125 mM NaCl, 25 mM NaHCO$_3$, 2.5 mM KCl, 1.25 mM NaH$_2$PO$_4$, 25 mM glucose, 2 mM CaCl$_2$, and 1 mM MgCl$_2$, equilibrated with 95% O$_2$ and 5% CO$_2$, ~320 mOsm. Experiments were performed in the absence of blockers except in a subset of EPSC experiments, in which gabazine (10 μM) was added to block inhibitory postsynaptic currents. We found no statistically significant differences between recordings in gabazine and in the absence of blockers (4 and 9 pairs, respectively) for EPSC$_1$ rise times (20–80%; 0.65 ± 0.18 ms versus 0.67 ± 0.06 ms, p=0.26), EPSC$_1$ decay τ (5.4 ± 0.97 versus 6.5 ± 0.54 ms, p=0.30), and total charge during the 3 x 50-Hz stimulus train (25.1 ± 10.1 versus 39.4 ± 7.6 pC; 60 ms integration time; p=0.31). These results are consistent with the assumption that GABAergic currents do not significantly contribute to evoked synaptic potentials in our recordings.

Presynaptic recording pipettes were fabricated from 2.0 mm / 0.6 mm (OD/ID) borosilicate glass tubing and had open-tip resistances of 10–20 MΩ. For tight-seal, bouton-attached stimulation under voltage-clamp conditions, the presynaptic pipette contained a K$^+$-based intracellular solution (140 mM KCl, 2 mM mgCl$_2$, 2 mM Na$_2$ATP, 10 mM EGTA and 10 mM HEPES), allowing us to measure bouton properties in a subsequent whole-bouton configuration. The presynaptic holding potential was set at –70 or –80 mV to minimize the holding current (typically between 0 and –5 pA) (*Alcami et al., 2012*; *Perkins, 2006*). Action potentials in mossy fiber boutons were evoked by brief voltage pulses (amplitude <1 V, duration 0.05–0.1 ms). Action currents showed an all-or-none

behavior, were reliably evoked by each short pulse above threshold, and followed the brief voltage stimuli after short time intervals (*Figure 1C*); late or spontaneous action currents did not occur (*Vyleta and Jonas, 2014*). Furthermore, action currents were tightly correlated with the occurrence of EPSCs in the recorded postsynaptic CA3 pyramidal neuron (*Figure 1C*). Thus, bouton-attached stimulation permitted the precise control of activity in a single presynaptic input. Boutons had diameters of ~3–5 µm, in close agreement with the previously reported range of diameters of mossy fiber boutons in light and electron microscopy studies (*Acsády et al., 1998*; *Amaral and Dent, 1981*; *Bischofberger et al., 2006*; *Chicurel and Harris, 1992*; *Rollenhagen et al., 2007*). In the whole-bouton configuration, capacitive currents decayed biexponentially, with a fast component corresponding to $C_1 = 1.17 \pm 0.46$ pF and a slow component corresponding to $C_2 = 3.16 \pm 0.82$ pF (seven pairs), further suggesting recording from typically sized boutons (*Amaral and Dent, 1981*; *Geiger and Jonas, 2000*). Stimuli (1 or three stimuli at 50 Hz) were delivered once every 20 or 40 s (0.05 and 0.025 Hz, respectively).

Postsynaptic recording pipettes were fabricated from 2.0 mm / 1.0 mm (OD/ID) borosilicate glass tubing with resistances of 1.5–3.5 MΩ and contained an internal solution with 130 mM K-gluconate, 20 mM KCl, 2 mM $MgCl_2$, 2 mM $Na_2ATP$, 10 mM HEPES, and 10 mM EGTA (pH adjusted to 7.28 with KOH, 312–315 mOsm). This formulation provides low series resistance for improved resolution of the kinetics of synaptic events and action potentials, and high electrode stability for accurately setting the resting potential of the cell, but sets $E_{Cl}$ at ~–45 mV. Because GABAergic signaling is hyperpolarizing at CA3 pyramidal neurons after early development (*Banke and McBain, 2006*), inhibition was not preserved in our recording conditions. In a subset of EPSC experiments, 2 mM QX-314 was added to block action potential initiation in the postsynaptic neuron. Current-clamp and voltage-clamp recordings were performed at –70 mV, and only recordings with <150 pA injection of hyperpolarizing current were included in analyses. Membrane potential was checked repeatedly throughout the experiment, and holding current was carefully readjusted if required. Postsynaptic series resistance ranged from 3.9–13.2 MΩ, median 5.9 MΩ. In current clamp mode, series resistance was fully compensated using the bridge balance circuit of the amplifier. In the voltage-clamp mode, series resistance was uncompensated, but carefully monitored with a test pulse following each data acquisition sweep. Only recordings with stable series resistance were included in analyses. Paired recordings with tight-seal, bouton-attached stimulation were stable for up to 30 min. EPSC data from four pairs had been included in a previous study (*Vyleta and Jonas, 2014*). Gabazine and QX-314 were from Biotrend, other chemicals were from Sigma-Aldrich and Merck.

## Data acquisition and analysis

Data were acquired with a Multiclamp 700A amplifier, low-pass filtered at 10 kHz, and digitized at 50 kHz using a CED 1401plus interface (Cambridge Electronic Design, Cambridge, UK). Pulse generation and data acquisition were performed using FPulse version 3.3.3 (U. Fröbe, Physiological Institute, University of Freiburg, Germany). Data were analyzed with Stimfit version 0.13.17 (*Guzman et al., 2014*) and Igor Pro (Wavemetrics). In a subset of recordings, traces of unitary EPSPs were filtered (low-pass digital filter, cut-off frequency of 1 kHz) for robust analysis of maximal slope of the rising phase of individual EPSPs. Mean EPSP and EPSC parameters were measured from average waveforms of individual recordings and averaged across recordings. Latencies were measured from the beginning of the presynaptic stimulus (0.1 ms duration) to the onset of the synaptic event unless stated differently. EPSP or EPSC onset was defined as the intersection of a line in the rising phase through 20 and 80% of the peak amplitude and the preceding baseline. For EPSPs, EPSCs, and action potentials, the analysis time window was set between 0 and 10 ms after the stimulus. EPSP decay time course was fit with a monoexponential function including an offset. For calculating facilitation ratios, EPSC peak amplitudes were normalized to the value for the first EPSC of a train. To quantify synaptic efficacy in EPSP recordings with overlaying action potential initiation in a subset of traces, the maximal slope of the EPSP was used as a parameter. The maximal EPSP slope was calculated as the maximum of the first derivative of the rising phase of the EPSP. Time-derivatives were calculated using a 60-µs temporal window. PTP decay time course was fit with a monoexponential function including an offset. To assess the extent of temporal summation, we simulated the superposition of three EPSPs of identical peak amplitude, shifted in time by 20 ms. The EPSP decay time constant was assumed to be 134 ms, consistent with the experimentally measured value for single unitary mossy fiber EPSPs (*Figure 1I*).

## Statistics and conventions

Statistical significance was assessed using a two-sided t-test at the significance level (P) indicated. Values are given as mean ± standard error of the mean (SEM). Error bars in the figures also represent the SEM; they were plotted only when larger than symbol size. Membrane potentials are reported without correction for liquid junction potentials.

## Acknowledgements

We thank Drs. Jozsef Csicsvari, Jose Guzmán, John Lisman, and Alessandro Treves for critically reading the manuscript. We also thank F Marr for technical assistance, E Kramberger for manuscript editing, as well as T Asenov (Miba machine shop) and M Schunn (preclinical facility) for efficient help. This project has received funding from the Austrian Science Fund (FWF P 24909-B24) and the European Research Council (ERC) under the European Union's Horizon 2020 research and innovation programme with grant agreement numbers 268548 and 692692 (to PJ).

## Additional information

### Funding

| Funder | Grant reference number | Author |
| --- | --- | --- |
| National Institutes of Health | 2T32HL083808 | Nicholas P Vyleta |
| People Programme | 291734 | Carolina Borges-Merjane |
| European Research Council | 268548 | Peter Jonas |
| Austrian Science Fund | FWF P 24909-B24 | Peter Jonas |
| European Research Council | 692692 | Peter Jonas |

The funders had no role in study design, data collection and interpretation, or the decision to submit the work for publication.

### Author contributions

NPV, Conception and design, Acquisition of data, Analysis and interpretation of data, Drafting or revising the article; CB-M, Analysis and interpretation of data, Drafting or revising the article; PJ, Conception and design, Analysis and interpretation of data, Drafting or revising the article

### Ethics

Animal experimentation: Animals were killed by rapid decapitation, in accordance with national, institutional, and European guidelines. Experimental procedures were approved by the Bundesministerium für Wissenschaft, Forschung und Wirtschaft (A. Haslinger, Vienna).

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
