## [Decision Letter]

Thank you for submitting your article "Plasticity-dependent, full detonation at hippocampal mossy fiber-CA3 pyramidal neuron synapses" for consideration by *eLife*. Your article has been reviewed by two peer reviewers, including Chris McBain (Reviewer #2), and the evaluation has been overseen by a Reviewing Editor and Eve Marder as the Senior Editor.

The reviewers have discussed the reviews with one another and the Reviewing Editor has drafted this decision to help you prepare a revised submission.

Summary:

This study explores the nature of transmission at the mossy fiber synapse between granule cells of the dentate gyrus and CA3 pyramidal cells in rat hippocampal slices in vitro. Although much is known about transmission at these synapses, e.g., low initial release probability and marked short term potentiation, the current paper re-explores the properties of transmission with high resolution recording techniques, combining cell attached recordings from individual presynaptic boutons with whole cell recordings from postsynaptic CA3 pyramidal cells. The authors demonstrate that single synaptic connections can be transformed from "conditional detonators" where reliable action potentials are triggered only after summation of a few synaptic potentials in a train, to one where the action potential probability triggered by unitary events is increased following high frequency induced post-tetanic potentiation. This transition converts synapses into reliable detonators.

Essential revisions:

The reviewers were uniformly enthusiastic about the work. One reviewer wrote, "This issue has major implications for how synaptic computations are performed in the granule cell-CA3 network and therefore is, in my opinion, of broad interest to the readership of *eLife*. Besides its significance, the data are of outstanding quality, which adds to the overall strength of the manuscript." Another referred to it as "a rather beautiful study," and said, "It is rare that I read a manuscript and have no comments on how it should be revised or improved. This is one of these manuscripts. I found the logic and flow of the experiments and data analysis clear and persuasive and have only one comment. The authors should be congratulated on providing such a lucid and straightforward dataset."

The main question had to do with the reversal potential for chloride used in the experiments and whether the extent to which it is expected to influence the results. The solutions as reported predict E_Cl_ around -45 mV (assuming complete dissociation), which could make IPSPs depolarizing and possibly excitatory (depending on where threshold is). While it is clear that PTP converts the response, it would be helpful to clarify (at least by discussion) any role for inhibition. The methods indicate that some experiments were performed with gabazine but it is not immediately evident which experiments these were. Therefore, for the revision:

1) Please verify that the solutions are reported correctly and indicate whether this is indeed the reversal potential.

2) Please also discuss the extent to which the results might be affected by inhibition. The specific queries of the reviewers are indicated in Major Point 1.

Major point:

1) The authors appear to have recorded the postsynaptic cells using pipettes filled with an intracellular solution that would be expected to set E_Cl_ at ~-45 mV, thus potentially turning feedforward GABAergic inhibition (triggered by the mossy fiber bouton stimulation) into excitation. Although this is not a problem for the single pulse experiment, as the authors show a very low probability of action potential generation in CA3 pyramidal cells, it does makes the interpretation of both the "three pulse" and "plasticity" experiments less unequivocal. In fact, one could think that repetitive activation of mossy fiber boutons might generate feedforward GABAergic excitatory responses, which would contribute to driving the pyramidal cell to action potential threshold. Also, previous work by the same lab has shown both PTP and LTP at the mossy fiber-dentate gyrus basket cells synapse. Therefore, it cannot be excluded that similar mechanisms might operate in the CA3 region and result in an enhanced excitatory feedforward GABAergic component (even to the first response) following the high frequency stimulation protocol. Are the authors sure that the composition of the intracellular solution given in the methods is correct?

---

## [Author Response]

The main question had to do with the reversal potential for chloride used in the experiments and whether the extent to which it is expected to influence the results. The solutions as reported predict E_Cl_ around -45 mV (assuming complete dissociation), which could make IPSPs depolarizing and possibly excitatory (depending on where threshold is). While it is clear that PTP converts the response, it would be helpful to clarify (at least by discussion) any role for inhibition. The methods indicate that some experiments were performed with gabazine but it is not immediately evident which experiments these were. Therefore, for the revision:

*1) Please verify that the solutions are reported correctly and indicate whether this is indeed the reversal potential.*

*2) Please also discuss the extent to which the results might be affected by inhibition. The specific queries of the reviewers are indicated in Major Point 1.*

Major point:

*1) The authors appear to have recorded the postsynaptic cells using pipettes filled with an intracellular solution that would be expected to set E_Cl_ at ~-45 mV, thus potentially turning feedforward GABAergic inhibition (triggered by the mossy fiber bouton stimulation) into excitation. Although this is not a problem for the single pulse experiment, as the authors show a very low probability of action potential generation in CA3 pyramidal cells, it does makes the interpretation of both the "three pulse" and "plasticity" experiments less unequivocal. In fact, one could think that repetitive activation of mossy fiber boutons might generate feedforward GABAergic excitatory responses, which would contribute to driving the pyramidal cell to action potential threshold. Also, previous work by the same lab has shown both PTP and LTP at the mossy fiber-dentate gyrus basket cells synapse. Therefore, it cannot be excluded that similar mechanisms might operate in the CA3 region and result in an enhanced excitatory feedforward GABAergic component (even to the first response) following the high frequency stimulation protocol. Are the authors sure that the composition of the intracellular solution given in the methods is correct?*

We confirm the use of a postsynaptic pipette solution with 20 mM KCl, which sets E_Cl_ at ~−45 mV. This solution was used for historic reasons in the lab, providing reduced series resistance for better resolution of the kinetics of synaptic events and action potentials, and improved electrode stability for accurately setting the resting potential of the cell. However, we appreciate the potential limitation of this approach, as pointed out by the reviewer. In the revised version, we have carefully discussed the possible effects of inhibition both with regards to (1) conditional detonation in response to trains of presynaptic stimuli and (2) full detonation in response to single APs in the PTP phase.

1) For the “three pulse” conditional detonation experiment (Figure 2), we agree with the reviewer that inhibition could make a contribution. If disynaptic GABAergic conductances contributed significantly to depolarization of the pyramidal neuron during presynaptic stimulus trains, we would predict that blocking GABAA receptors with gabazine would significantly change the time course of postsynaptic currents and reduce the total charge during train stimulation (Figure 2). We have tested this prediction in our experimental data. However, we found no statistically significant differences between recordings in gabazine (10 µM) and in the absence of blockers (n = 4 and 9, respectively) for EPSC_1_ rise times (20-80%; 0.65 ± 0.18 ms versus 0.67 ± 0.06 ms, P = 0.26), EPSC_1_ decay τ (5.4 ± 0.97 versus 6.5 ± 0.54 ms, P = 0.30), and total charge during the 3 x 50-Hz stimulus train (25.1 ± 10.1 versus 39.4 ± 7.6 pC; 60 ms integration time; P = 0.31). These results are consistent with the assumption that GABAergic currents do not significantly contribute to evoked synaptic potentials in our recordings. Importantly, the “conditional detonation” we observe in response to stimulus trains is very similar to that previously reported in in vivo experiments where inhibition is fully intact (Henze et al., 2002). We have added the corresponding quantitative data to the Methods section (subsection “Bouton-attached stimulation and paired recording”). We have also added a statement to the Methods section describing that inhibition was not preserved in our recording conditions, and have included a reference to Banke and McBain (2006), where GABAergic signaling was shown to be hyperpolarizing at CA3 pyramidal neurons after early development (subsection “Bouton-attached stimulation and paired recording”, last paragraph).

2) For the “tetanus” full detonation experiment (Figure 3), we think that a confounding effect of inhibition is highly unlikely, because the onset of feedforward inhibition is too slow (Torborg et al., 2010). In our experiments, the latency from the stimulus to the maximum slope of the EPSP preceding AP generation in the PTP phase was 1.5 ± 0.09 ms (n = 7). Torborg et al., (2010) determined that the latency between monosynaptic excitation and disynaptic inhibition in CA3 pyramidal neurons was, on average, 2.2 ms. Thus, feedforward inhibition would be predicted to occur ~3.7 ms after the stimulus. However, in our experiments the latency from the stimulus to action potential onset in the PTP phase was only 3.1 ± 0.09 ms (n = 7). Thus, disynaptic inhibition comes too late to affect the probability of action potential initiation. Consistent with this argument, Torborg et al., (2010) found no effect of blocking GABAA receptors on latency, jitter, and initiation probability of single action potentials evoked by mossy fiber stimulation, and concluded that “[…] mossy fiber-driven feedforward inhibition does not substantially influence the timing of single action potentials in CA3 pyramidal cells, but rather provides a potent mechanism to prevent excessive depolarization and burst firing of CA3 pyramidal cells.” Furthermore, the authors concluded that “[…] the temporal precision of action potentials is likely governed by the kinetics of the EPSP itself […] largely independent of inhibition” (Torborg et al., 2010). We have added a sentence to the Results section to describe the latency data (last paragraph). We have also added a paragraph to the Discussion to explain the role of inhibition, including a reference to the paper by Torborg et al., 2010 (third paragraph).